# The Impact of CEO Characteristics on the Financial Performance of Family Businesses Listed in the Euronext Exchange

Zouhour El Abiad [1], Rebecca Abraham [2,*], Hani El-Chaarani [3], Yahya Skaf [4], Ruaa Omar Binsaddig [5] and Syed Hasan Jafar [6]

1    Department of CRED, ESA Business School, Beirut 289, Lebanon; esa@esa.edu.lb
2    Department of Finance and Economics, Nova Southeastern University, Fort Lauderdale, FL 33328, USA
3    Department of Management, Beirut Arab University, Beirut P.O. Box 11-5020, Lebanon; h.shaarani@bau.edu.lb
4    Department of Accounting and Finance, Notre Dame University-Louaize, Zouk Mosbeh 5425, Lebanon; yskaf@ndu.edu.lb
5    Department of Finance, University of Business and Technology, Jeddah 23435, Saudi Arabia; r.binsaddig@ubt.edu.sa
6    Department of Finance, Woxsen University, Telangana 502345, India; syedhasan.jafar@gmail.com
*    Correspondence: abraham@nova.edu

**Abstract:** This paper identifies the CEO characteristics that have an impact on the performance of family businesses listed in the Euronext in the post-COVID 19 period. CEO characteristics are evaluated on two dimensions, i.e., personal characteristics and corporate governance mechanisms. A sample of 137 firm-year observations from Portugal, Luxembourg, the Netherlands, Ireland, France, and Belgium was chosen. CEO attributes of age, gender, education, and family membership were combined with corporate governance mechanisms of ownership concentration, CEO duality, CEO directorships, and CEO tenure, to predict return on assets and return on equity, using OLS regression. GMM estimation and Two-Stage Least Squares were employed to establish the robustness of the results. Among CEO personal characteristics, CEO family membership has a positive impact on return on assets, and a positive impact on return on equity. Among corporate governance mechanisms, CEO duality had a negative impact on return on assets, and a negative impact on return on equity. CEO ownership, and CEO tenure had a positive impact on return on assets, and a positive impact on return on equity. This paper's value lies in its evaluation of the under-researched area of family businesses of Euronext-listed firms. It can be used by family businesses in the region, for the selection and training of CEOs to fulfill the goal of achieving superior financial performance.

**Keywords:** CEO characteristics; family business; Euronext region; financial performance; CEO duality; CEO tenure

## 1. Introduction

Family businesses are unique entities. They are frequently composed of a founder with offspring from successive generations, in an array of cousins, aunts, uncles, and in-laws. Loyalty to the family is the unifying value, which is demonstrated in varying degrees by diverse family members. Family members may or may not be employed in the family business. Family businesses may range from small and medium-sized startups to giant conglomerates in the third or fourth generation. As an example, Mars is a large privately owned family business with partnerships in multiple locations, spanning a century. This paper defines family businesses as family owned. We use Klein's (2000) classification, which distinguishes between family-owned firms and family-governed and family-managed firms. In family-owned firms, successive generations of founder siblings, cousins, and their descendants have large equity stakes. This form of family business differs from others in the Klein (2000) categorization, which are family businesses by governance or management.

Family ownership is marginal with family members serving on supervisory boards or management boards.

The literature provides some support for superior ownership and control displayed by family-owned businesses. The family's large undiversified equity ownership permits it to safeguard against managerial expropriation (Demsetz and Lehn 1985). The family's long investment horizon is apparent in the desire to sustain the business in the foreseeable future, as the business is the chief source of intergenerational wealth transfer (James 1999). As leaders, CEOs of family businesses drive strategic decision making. They set the direction for the firm in terms of products, processes, and markets. They inspire the top management team, and maintain cordial relations with the board. They regularly channel information to family members, involving them in different aspects of the business, to obtain their ongoing support. Anderson and Reeb (2003) observed that the presence of the CEO founder and CEO descendants significantly increased profits as measured by return on assets in a sample of U.S. family businesses. Further, they observed that equity-based compensation for the CEO significantly increased both return on assets and firm value.

A review of existing studies reveals a research gap in measuring CEO attributes on the financial performance of family businesses. The coverage of CEO attributes in family businesses in the literature has been sparse. Gottesman and Morey (2010) found a link between education and financial performance in family firms based in the United States, with graduate degree holders having a positive effect on return on assets and return on equity. Among Indian respondents, long-tenured female CEOs negatively influenced financial performance (Kaur and Singh 2019). Saidat et al. (2022) evaluated the CEOs of Jordanian banks, observing that, as members of the family, they were cognizant of family dynamics. Family membership also increased commitment to the firm's success. Li et al. (2007) examined family businesses in China in the immediate aftermath of privatization. CEOs with large ownership stakes had a positive impact on financial performance. The purpose of this study is to close the research gap of limited empirical examinations of CEO characteristics on family firm performance in the context of theories of CEO attributes explaining firm performance. We select the Euronext region as the source of CEOs in family businesses.

Upper Echelon Theory, as presented by Hambrick and Mason (1984), may provide the justification for the importance of CEO attributes in explaining firm performance. It sets forth that significant judgements such as setting a strategic direction for the firm, may be based on the CEO's personal characteristics, such as age, gender, nationality, and tenure. While Upper Echelon Theory may explain CEO characteristics in both family firms and non-family firms, Stewardship Theory is particularly applicable to family businesses. CEOs perceive themselves as stewards of the family business, being charged with taking actions that promote the best interests of the business in terms of setting a strategic direction, inspiring employees and positioning resources for maximum profitability. Donaldson and Davis (1991) theorized that such stewardship results in superior financial performance.

This study advances knowledge in multiple ways. First, it supplements U.S.-based family business studies with a non-U.S. sample. As the literature on family businesses is centered in the United States (see Anderson and Reeb 2003, for a review), a non-US region is used. Second, it examines the effects of both personal characteristics and corporate governance mechanisms on firm performance in a unique framework. The first dimension is the personal characteristics of the CEO, such as age, gender, and education. Intuitively, values and predispositions are a function of these demographic characteristics, guiding the CEO's managerial decisions, to hire talent, rationally evaluate alternative courses of action in product selection and market expansion, and labor union negotiations. CEO education, gender, and nationality have been found to explain firm performance in publicly traded firms (Kaur and Singh 2019). Kaur and Singh (2019) cite the following studies. Western and Asian studies have found opposing effects for gender, with female CEOs outperforming males in certain studies (Brennan and McCafferty 1997; Peni 2014), and male CEOs outperforming female CEOs in other studies (Amran 2011). CEO education has been

shown to benefit corporate risk taking, and in turn, financial performance (Farag and Mallin 2016; Wang et al. 2016). The nationality of the CEO supports financial performance in certain cases (Badru and Raji 2016), or fails to be associated with firm performance in other samples (Huang 2013; Rivera and DeLeon 2005). As none of these studies employ samples from family businesses, we wish to rectify this gap in the literature. The second dimension of the effect of CEO characteristics on firm performance depends upon the corporate governance mechanisms prevalent in the environment within which the CEO operates. A key element is ownership concentration. As family members are large stakeholders, their high ownership concentration could lead them to rigorously monitor and evaluate CEOs. Empirically, mixed results have been obtained in non-family businesses, suggesting that agency costs may moderate CEO performance's relationship with high ownership concentration, in that CEOs who are large shareholders may invest in low-NPV projects, which adversely affect firm performance (Morck et al. 1988). In the absence of agency costs, ownership concentration was found to positively influence firm performance (Kaur et al. 2013). CEO tenure can have a positive effect on firm performance in its ability to build loyalty and commitment (Vintilă et al. 2015), or a negative effect on firm performance if long-tenured CEOs become insular and incapable of creative problem solving.

Third, these two dimensions of the effect of CEO characteristics on the performance of family businesses must be evaluated in a region rich in family businesses. Such a region may be found in the five countries that make up the Euronext exchange. Euronext operates regional exchanges in Belgium, France, Ireland, Italy, the Netherlands, Norway, and Portugal. It has a strong tradition of family businesses (Dana et al. 2022), with leading companies being family-owned and operated (consider LVMH and L'Oreal as examples). Secondly, the existing literature on firm performance for firms listed in the Euronext does not directly address CEO attributes. For instance, in successive studies, Vieiria (2017) observed that economic adversity and debt policy influenced the financial performance of family businesses in Portugal. Likewise, Teodosio et al. (2022) found that board size, age of the board, and number of independent directors increased systemic risk in a similar sample. The corporate governance mechanisms in the latter study did not include the CEO's governance activities, such as CEO duality and CEO ownership concentration. A few studies of Euronext countries include non-Euronext countries (Cucculelli et al. 2019; Wasowska 2017), so that it is unclear if measures of CEO activity can be unbiasedly attributed to CEOs of family businesses in countries listed in the Euronext exchanges.

Variable selection may be explained as follows. The first category of characteristics were demographic variables, including gender, age, education, and family membership. Variations in these variables justified their inclusion. Gender was included as there may be a gender difference in risk taking, with women being less inclined to take risks than men, as observed by Kaur and Singh (2019). Risk taking may be essential for family businesses as it encourages the development of new products and expansion into new markets. Age was included as older CEOs may be able to secure supplier agreements and distribution channels due to longstanding relationships with suppliers and distributors. Younger CEOs do not have this advantage. Education confers financial expertise on certain CEOs, enabling them to comprehend financial results about profits, debt, and firm value over counterparts who lack this knowledge. Corporate Governance mechanisms influence CEO conduct differentially, creating another category of variables to be measured. CEO ownership concentration consists of CEOs who are major shareholders who demand accountability for financial results. CEO duality may result in CEOs failing to objectively evaluate top management's performance as they are part of top management. CEOs with multiple directorships acquire a range of skills beyond others who serve on a single board. Tenured CEOs may display a level of loyalty and familiarity with the firm that others have not acquired.

To avoid any confounding of results due to lockdowns of businesses due to COVID-19, we confine our investigation to the post-COVID-19 period. While this restriction limits the

period under examination, it does ensure that adverse effects on firm performance from the closure of businesses from 2020 to 2021 have been eliminated.

The remainder of this paper is organized as follows. Section 2 is a review of the literature and hypotheses development, Section 3 is methods, Section 4 is results, Section 5 is discussion, and Section 6 is conclusions.

## 2. Literature Review and Hypotheses Development

In the literature review, we explore the background literature for hypotheses development. As the impact of CEO characteristics on firm performance has largely been examined in non-family businesses, we commence with a description of these findings in Section 2.1. Then, we contextualize CEO characteristics in theories in Section 2.2. Finally, we describe the existing literature on CEO family businesses in Europe, the region of our study. This approach first creates the framework for the effects of particular characteristics, both empirically and theoretically, then shows the existing studies pertaining to family businesses in the Euronext region to reveal research gaps to be closed by this study. The remaining sections develop the hypotheses to be empirically tested in this study.

### 2.1. CEO Charateristics That Influence Firm Performance in Non-Family Businesses

A wealth of literature has examined CEO characteristics in terms of their impact on firm performance in non-family businesses. We consider these studies as we wish to extend their investigations to family businesses. The following characteristics are considered.

CEO Age: Both positive and negative effects were observed. Peterson et al. (2001), Huang et al. (2012) and Belot and Serve (2018) observed that older CEOs positively influenced financial reporting quality. Conversely, using a sample of Indonesian firms, Suherman et al. (2023) observed that older CEOs were conservative and averse to the necessary risk taking needed to explore new products and markets. Consequently, their firms underperformed.

CEO Gender: Espinosa-Mendez and Correa (2022) used a Chilean sample to observe superior financial performance in SMEs led by female CEOs. A similar result was obtained by Suherman et al. (2023) for an Indonesian sample, with female CEOs providing objectivity in the assessment of management. A plethora of earlier studies concurred with Kotiranta et al. (2007), Adams et al. (2009), and Ng (2017). A more indirect effect was observed by Pandey et al. (2023) who found that the presence of female CEOs reduced the adverse effects of CEO duality on firm performance. Female CEOs introduced objectivity into the board's evaluation of top management in firms with the CEO serving as the Board Chair.

Education: In an early study, Ng and Feldman (2009) linked CEO educational backgrounds to innovation and strategic change. Educated CEOs pursued innovations that brought about long-term strategic improvements in firm performance. Suherman et al. (2023) found that educated CEOs were able to meet the intellectual needs of their jobs, thereby increasing return on assets, return on equity, and Tobin's q.

Tenure: Francis et al. (2008) found that tenured CEOs used their decades-long familiarity with the firm to easily correct errors in financial reports, thereby enhancing financial reporting quality. Their alignment with teams and strong links with internal business assist in resolving conflicts, thereby improving financial performance (Ali and Zhang 2015).

CEO Duality: Palaniappan (2017) used an Indian sample to show that the lack of objectivity in evaluating top management performance of firms by Board Chairs who are themselves CEOs or members of top management reduces financial performance. Similar results have been obtained by Pandey et al. (2023) for another Indian sample.

To this mix of characteristics, we wish to add CEO ownership concentration, directorships, and family membership in our assessment of family businesses. In family businesses, ownership concentration means that the CEO is a large shareholder who can reduce agency costs. CEOs with multiple directorships bring a variety of skills and experiences to family businesses, which may not have access to these resources. Family membership could

have a positive effect on firm performance as CEOs care for the family business which is their inheritance.

*2.2. Theories of the Effect of CEO Characteristics on the Financial Performance of Family Businesses*

Upper Echelon Theory (Hambrick and Mason 1984) sets forth that managerial characteristics can be useful predictors of organizational outcomes. CEO behavior is shaped by their age, gender, tenure, education, and nationality. These demographic attributes expose them to experiences that create a knowledge base of experiences and values that influence their interpretation of corporate situations. For example, a CEO who has grown up in large city of a major industrialized nation has been exposed to individuals from diverse cultures, with a range of skills. Such a CEO may be more open to considering future employees from diverse cultures and socio-economic backgrounds in hiring decisions. Thus, CEOs make decisions based on their interpretations of situations based on their background and characteristics. Behavioral Finance maintains that psychological biases, either due to personal preferences or misconceptions, may adversely affect the quality of decision making, and in turn, result in suboptimal financial performance (Baker and Wurgler 2013; El-Chaarani 2014).

The Resource Dependency Theory links with the Upper Echelon Theory by relating CEO characteristics to the resources and needs of the organization. Pfeiffer and Salancik (1978) proposed the Resource Dependency Theory which views CEO appointments as being based on their skills, potential, and experience. These attributes must be effectively matched to the resources and needs of the organization to ensure effective management and effective organizational performance. To revisit the example of the CEO who may be attracted to employees from diverse backgrounds, the position for which the employee is being considered may involve coordinating with others in a cross-functional team. The CEO's broad perspective matches the needs of the position in an application of Resource Dependency Theory. In family businesses, resources are often scarce, as the businesses do not have the corporate relationships required to secure capital and purchase inputs into the production process. The appointment of a CEO, who either has power in bargaining relationships or can locate individuals with such power, can offer a competitive advantage in the acquisition of resources.

Corporate governance mechanisms may be considered to be the internal structures to which CEO characteristics are matched to achieve superior performance. This may be particularly true for family businesses in which the CEO's are able to create an environment of accountability and satisfaction among employees, balanced with the family's needs for stability and continuity of the revenue stream that has historically sustained financial performance. We consider CEO ownership, CEO duality, and CEO directorships as corporate governance mechanisms that influence the financial performance of family businesses. Stewardship Theory supports the CEO perceiving the self as the steward of the family's interests, motivated by the desire to further the family business's financial performance. If the CEO is a family member, he or she may own sufficient stock so as to be adversely affected by poor financial performance. This outcome may motivate the CEO to adopt strategies that are crucial for success. If the CEO is a non-family member, stock ownership may mitigate agency conflicts. As the owner, the CEO may advance financial performance to advance personal financial interests. CEO duality, whereby the CEO is the Chairperson of the Board of Directors, may increase management's accountability. As the Board Chair, the CEO may prevail upon directors to closely monitor the performance of managers about whom the CEO has in-depth information, as the CEO is also a member of the management team. CEOs who serve on multiple boards acquire broad knowledge of multiple industries and organizational cultures. This knowledge enables them to effectively manage inter-organizational relationships (Geletkanycz and Boyd 2011).

### 2.3. CEOs of European Family Businesses

We examine a few studies of CEO characteristics of family firms in Europe, rather than exclusively in the Euronext, which has a paucity of empirical studies. The results are mixed, with relational capital benefitting family CEOs, while CEO duality and internationalization support the presence of non-family CEOs. Cucculelli et al. (2019) observed that strong relationships between family CEOs and banks led to increased success in obtaining loans. Non-family CEOs were significantly less successful than family CEOs in securing loans in seven European countries, of which only two were listed in the Euronext exchange. In Southern Europe, family involvement had no effect on CEO duality and firm performance, suggesting that in a few of the Euronext countries, non-family CEOs may be as effective as family CEOs in achieving superior performance if they hold the dual roles of CEO and Chairperson of the Board of Directors (Garcia-Ramos and Garcia-Olalla 2014). In studies of internationalization of family firms in a single non-Euronext country (Westhead and Howorth 2006) and a sample dominated by five non-Euronext countries (Wasowska 2017), family CEOs had reduced propensity to export over non-family CEOs.

### 2.4. Personal Characteristics of CEOs and Family Firm Performance

#### 2.4.1. CEO Gender and the Financial Performance of Family Firms

Nekhili et al. (2018) set forth that men espouse transactional leadership styles. The transactional leader views relationships as transactions, whereby a beneficial action by one party is reciprocated. If there is no beneficial action, the relationship is terminated. In a family business focused on increasing family wealth, transactional leaders will only engage with suppliers and customers if the suppliers continue to supply inputs, and customers continue to purchase products. Over time, successive inputs into production, and repeat customers will contribute positively to firm performance (El Nemar Sam et al. 2022).

We suggest that women are unlikely to be transactional leaders. Their relationships with suppliers, customers, employees, and family members are likely to be nurturing and supportive, with the view to sustaining long-term relationships. Dissatisfaction with suppliers, customers, or family members will rarely terminate relationships. Instead, women will attempt to resolve conflicts, possibly when relationships do not yield financial benefits to the family firm. Retaining the aunt, uncle, or cousin with whom there is a disagreement as an ally may be more important than terminating a relationship that is detrimental to the family business's performance.

Khan and Vieito (2013) theorized that women may have lower risk tolerance than their male counterparts. Intuitively, risk taking is needed to embark upon new projects, create new products, and hire employees with special talents. Many family firms are insular, in that they were started by a founder a century ago, and have embarked on a long history of preservation of intergenerational wealth. These firms need to take risks by exploring new paths, while creating new products. If women CEOs are excessively risk-averse, creativity will fail to materialize, and performance will stagnate. Successive empirical studies have found that male CEOs of non-family firms have produced significantly higher stock returns than female CEOs (Amran 2011; Strelcova 2004).

**Hypothesis 1.** *The presence of women as CEOs of family businesses may be associated negatively with firm financial performance.*

#### 2.4.2. CEO Age and the Financial Performance of Family Businesses

Peni (2014) maintains that the experience and quality of management supported by age improves financial performance. In the context of family businesses, a senior CEO has a plethora of external relationships, with government, industry groups, and other stakeholders to assist the business in coping with uncertainties. Most of the European family businesses are small and medium-sized (SMEs) businesses. A sudden change in regulations regarding packaging, labeling, and exporting products could leave a youthful

CEO unsure of the measures to be taken. A senior CEO is at the center of a web of longstanding relationships. This senior CEO could call upon trade partners, attorneys, and regulators, whom they have built relationships with over time. Their advice may be sufficient to resolve the matter. This suggests that older CEOs have the ability to ensure the stability and continuity of the family business enterprise. Such stability and continuity is essential for the intergenerational wealth transfer that forms the basis for the continuity of the family business into successive generations.

Hirshleifer (1993) observed that young CEOs were more focused on achieving short-term goals. In an examination of banks, young CEOs were able to boost short-term financial performance. However, if the goal is shareholder wealth maximization, temporary increases in short-term financial performance may result in higher return on assets, or higher return on equity for a few quarters, with no significant positive effect on long-term finance. For a family business that seeks to sustain a multitude of aunts, uncles, cousins, and in-laws, stable long-term growth is desired, as it provides financial sustenance in the distant future. Therefore, the short-term focus of young CEOs runs counter to the long-term financial health of the family enterprise.

**Hypothesis 2.** *There is a positive relationship between CEO age and family firm financial performance.*

2.4.3. CEO Educational Level and the Financial Performance of Family Firms

We set forth that formal education imposes self-discipline in the sense that a student must submit assignments, take exams, and meet with professors at appointed dates and times. Creative projects, critical thinking assignments, and case analyses require the use of higher-level analytical skills. Masters theses and doctoral dissertations require the theoretical development of an original problem, followed by its statistical validation. Multiple higher-level skills are deployed, including judgement, the ability to synthesize and integrate multiple streams of thought, followed by the discernment to select the most appropriate method of statistical analysis. In family firms, where there is less specialized expertise to provide analytical problem solving, it is incumbent on the CEO to demonstrate such attributes. This suggests that in family firms, CEO educational level, particularly at the post-graduate level, confers the cognitive skills and capabilities needed for successful financial performance.

This is the position taken by Goll et al. (2007), who observed superior problem solving capabilities in complex situations by CEOs with graduate degrees, albeit in non-family firms. Such CEOs displayed optimism by placing more emphasis on opportunities rather than threats (Karami et al. 2006).

Yet another attribute of CEOs pertaining to educational level is financial expertise, which has been shown to result in superior financial reporting (Gupta and Mahakud 2020) and earnings management (Gounopoulos and Pham 2016), which is turn, lead to superior financial performance in non-family firms. In a family firm, a finance background can be valuable in that the CEO can pinpoint divisions/areas of financial weakness, identify the financial concerns of regulators, and take corrective action upon observing increases in expenses or shortfalls in revenue.

**Hypothesis 3.** *There is a positive relationship between CEO educational level and family firm performance.*

2.4.4. CEO Family Membership and Family Firm Financial Performance

We conjecture that family membership provides an impetus for the adoption of strategies by the CEO that improve family firm performance. In the Euronext region, the small size of a family firm results in close relationships among family members. Each member of the family has a stable position within the family hierarchy. The status within the family confers a sense of security, so that family members are committed to maintaining their position within the family. The feelings of belongingness forge an emotional connection between the CEO and his or her family members. Such emotional attachment spurs the

CEO to make prodigious effort to improve firm family performance. Empirically, in successive studies of family firms outside of the Euronext region, the presence of family CEOs was associated with higher levels of return on assets, return on equity, and Tobin's Q (Fahlenbrach 2009; Kowalewski et al. 2010).

**Hypothesis 4.** *There is a positive relationship between family membership of CEOs and family firm financial performance.*

*2.5. Corporate Governance Mechanisms and Financial Performance*

2.5.1. CEO Ownership and Family Firm Performance

Agency conflicts are eliminated if the CEO is a large shareholder. Jensen and Meckling (1976) set forth agency theory as explaining the conflict between shareholders (owners) and management. Shareholders pursue the goal of wealth maximization through improved financial performance. In contrast, managers may pursue negative NPV projects that yield private benefits. If the CEO is a large shareholder, he or she derives no benefit from pursuing initiatives that are detrimental to financial performance, as such actions will reduce the value of their security portfolios. Therefore, CEOs of family firms are motivated to increase the family's wealth by forming industry partnerships, creating new products, and seeking new markets. El-Chaarani et al. (2022) observed that the presence of large shareholders of banks in the Middle East and North Africa region on the boards of banks resulted in increased return on assets and increased return on equity. Likewise, ownership concentration on bank boards predicted financial performance in the Gulf Cooperation Council countries (El-Chaarani et al. 2023). Even though these empirical studies were conducted in non-family banks, the elimination of the separation of ownership and control that is the basis for agency conflicts is similar to family firms. The presence of large shareholders on the board of directors gave the board a personal stake in promoting financial performance goals.

**Hypothesis 5.** *CEO share ownership has a positive effect on family firm performance.*

2.5.2. CEO Duality and Family Firm Performance

The sample consists of boards with a one-tier governance structure, consisting of a Board Chair who may or may not be a member of top management and directors who are both managers and independent directors. We employ the one-tier board structure definition contained in the OECD 2023 Factbook. CEO duality is the joint appointment of a single individual to the positions of CEO and Chairperson of the Board. Stewardship theory posits that CEOs are caretakers of the organization, being imbued with a feeling of responsibility toward customers, suppliers, employees, and members of the management team. We infer that CEOs as stewards will use cooperative relationships to further the interests of the organization. In family businesses, CEOs balance family interests with those of the other stakeholders by achieving compromises that leave all parties in the conflict with some cause for satisfaction. For example, a family member may wish to hire a personal friend in a senior position for which the individual is unqualified. The dual CEO–Chair, may point out that this person is better suited to another position. By gaining the support of the other members of the board, as well as the top management team, the dual CEO–Chair may be able to convince the family member of the friend's suitability for an alternative position. The links that the CEO has with board members may facilitate an acceptable choice. In another example, a single division of the family enterprise may show declining performance. The need for accountability on the part of the management of that division may be realized by a dual CEO. The dual CEO could obtain the Board's support for impartial evaluation of the management of the division, if the CEO leads the board. If the CEO is not the Board Chair, supporters of the management of the failing division may prevail upon the board to relax actions to hold management accountable.

They may distract the board from objective reviewing of the managers by claiming that external circumstances are responsible for the division's failure.

**Hypothesis 6.** *CEO duality has a positive effect on family firm performance.*

### 2.5.3. CEO Directorships and Family Firm Performance

We set forth the following argument. Family firms, particularly those of small size, have limited exposure to new ideas, modern production methods, creative methods of funding, and the advice of experts in logistics, marketing, and human resource management. If the CEO holds multiple directorships, he or she would have access to firms in which the aforementioned novel strategies are being employed. The CEO may interact with the board members of other firms, who have access to the knowledge and resources that could be applicable to the needs of his or her family business. As family relationships form the core relationships of family businesses, CEOs may historically have strived to build and maintain such relationships. Therefore, a CEO who has multiple directorships can infuse an insular family firm with novelty, excitement, and energy, derived from creating external partnerships, which lead to future growth opportunities.

Geletkanycz and Boyd (2011) observed the benefits of increased knowledge, improved monitoring, and useful advice derived by CEOs with multiple directorships in non-family firms, leading to improved firm financial performance.

**Hypothesis 7.** *There is a positive relationship between CEO directorships and family firm performance.*

### 2.5.4. CEO Tenure and Family Business Performance

CEO tenure is the time a CEO stays in the position. Peni (2014) set forth that experienced leaders with a long history at a firm have the advantage of developing a deep understanding of the business, along with developing a plethora of relationships with customers, suppliers, regulators, and other stakeholder groups.

We present the following reasoning. In the context of family businesses, the depth of understanding of multiple departments, different products, and roles of family members in leading different segments of the business is particularly valuable in forming rational judgements about the strategic direction of the firm. As an example, a century-old tire manufacturing family business which exports to multiple countries may benefit from a tenured CEO who remembers the addition of each export market over time. The CEO may be able to prevail upon the heads of growing export divisions to increase sales, sell new products, and advance growth. The CEO may be able to win family support for these new initiatives, using friendships with key family members built over time. Suggestions for new divisions may emerge from these stakeholders, based on their observations of industry trends. The stakeholders may feel free to discuss the creation of these new departments with CEOs with whom they have built longstanding relationships. In a smaller family business, such as a bakery, family members may form most of the staff. The seniormost member of the family may be the CEO, whose decades of association with family members with specialized skills provides knowledge of their likes and dislikes. By supporting the personal preferences of these family members, the CEO will create the warm, nurturing environment in which they feel secure, thereby increasing labor productivity.

**Hypothesis 8.** *CEO tenure is positively associated with the financial performance of family businesses.*

### 3. Methods and Materials

*3.1. Data Collection and Sample Characteristics*

The data were collected from listed companies in the Euronext Exchange in 2023. Firms qualified for inclusion in the sample based upon the criteria of (1) family ownership, (2) succession by family members, and (3) the presence of family members in the position of CEOs, or family membership on the Board of Directors. The resulting sample consisted of

85 firms. The financial and non-financial data were collected from annual reports, Euronext, and Datastream databases. Fourteen family firms were excluded, due to missing data. The final sample consisted of 71 family firms, with 137 firm/observations. Firm locations included Portugal (1 firm), Netherlands (2 firms), Luxembourg (1 firm), Ireland (1 firm), France (55 firms), and Belgium (11 firms).

Descriptive statistics of 137 firms/observations in 2021 and 2022 are shown in Table A1. In total, 89% of CEOs were males, with an average age of 55.53 years and 8.84 years of tenure. Educational levels were high, with 68.82% of the CEOs having graduate degrees, special diplomas, or certificates. The majority of CEOs (57.29%) were family members, who owned 8.84% of the family firm listed in the Euronext Exchange. They had two board seats on the boards of other organizations. CEO duality was common, as 69.57% of firms had CEOs who simultaneously served as Board Chairs.

Table 1 shows the means and standard deviations of the family firms listed in the Euronext Exchange.

**Table 1.** Descriptive statistics of all variables.

| Symbol | Item | Average | Standard-D | Minimum | Maximum |
|--------|------|---------|------------|---------|---------|
| ROA | Return on Assets | 0.1866 | 0.0482 | 0.1209 | 0.3191 |
| ROE | Return on Equity | 0.107498 | 0.042991 | 3.0183 | 16.002 |
| GEN | Gender | 0.8932 | 0.1891 | 0 | 1 |
| AGE | Age | 55.5384 | 7.7843 | 6.2021 | 41.0491 |
| EDU | Education | 0.6882 | 0.0996 | 0 | 1 |
| FAM | Family Membership | 0.5729 | 0.5001 | 0 | 1 |
| OWN | Ownership | 8.8491 | 9.9889 | 5.2339 | 18.0482 |
| DUA | Duality | 0.6957 | 0.1483 | 0 | 1 |
| DIR | Directorship | 2.5013 | 0.3282 | 0 | 4 |
| TEN | Tenure | 8.8467 | 3.3178 | 3.2031 | 19.4041 |
| FAGE | Firm Age | 2.00912 | 9.8471 | 8.9481 | 45.3021 |
| FSIZE | Firm Size | 8.3747 | 1.4721 | 6.3991 | 12.4945 |

*3.2. Definition of Variables*

Table 2 describes the dependent variables, independent variables, and control variables used in the study.

**Table 2.** Definition of variables.

| Variable | Type of Variable | Description of Variable |
|----------|------------------|-------------------------|
| Return on Assets | Dependent | Net income/Total Assets |
| Return on Equity | Dependent | Net income/Stockholders' Equity |
| Gender | Independent | Dichotomous, 0 for men, 1 for women |
| Age | Independent | Number of years |
| Education | Independent | Dichotomous, 0 = undergraduate degrees, 1 = graduate degrees, or graduate certifications |
| Family Membership | Independent | Dichotomous, 1 = CEOs who are family members, 0 = CEOs who are non-family members |
| Ownership Concentration | Independent | Number of shares owned by the CEO/Number of shares outstanding |
| CEO Duality | Independent | Dichotomous, 1 = Presence of CEO Duality, 0 = Absence of CEO Duality |
| Directorships | Independent | The number of board positions held by the CEO |

**Table 2.** *Cont.*

| Variable | Type of Variable | Description of Variable |
| --- | --- | --- |
| Tenure | Independent | The number of years a CEO stays in their current position |
| Firm Age | Control Variable | The logarithm of the number of years since the firm was established |
| Firm Size | Control Variable | The natural logarithm of total assets (thousands of EUR) |

*3.3. Data Analysis*

OLS regression was employed to test the hypotheses, followed by GMM estimation and Two-Stage Least Squares to test the results for robustness. The following equations were specified,

$$
\begin{aligned}
Returnonasset ={}& \alpha + \beta_1 Gender + \beta_2 Age + \beta_3 Education + \beta_4 Family\ Membership + \beta_5 Ownership \\
&+ \beta_6 Tenure + \beta_7 Directorships + \beta_8 CEO\ Duality + \beta_9 Firm\ Age + \beta_{10} Firm\ Size + \varepsilon_1
\end{aligned}
\tag{1}
$$

$$
\begin{aligned}
Returnonequity ={}& \alpha + \beta_1 Gender + \beta_2 Age + \beta_3 Education + \beta_4 Family\ Membership + \beta_5 Ownership \\
&+ \beta_6 Tenure + \beta_7 Directorships + \beta_8 CEO\ Duality + \beta_9 Firm\ Age + \beta_{10} Firm\ Size + \varepsilon_2
\end{aligned}
\tag{2}
$$

As French firms constituted 77% of the sample, the analysis was repeated for sub-samples of French firms, and non-French firms to detect if the full sample was biased in the direction of French firms. No bias was detected as results from the sub-samples matched those from the full sample.

## 4. Results

Table 3 is a correlation matrix. Table 4 shows the results of the regressions of return on assets and return on equity on CEO characteristics, used to test Hypotheses 1–8. Hypothesis 1 was not supported, as CEO gender had no significant impact on return on assets (coefficient = 0.1034, $p > 0.05$) or return on equity (coefficient = 0.1504, $p > 0.05$). However, this result may change as the sample size is small and chiefly male. A larger sample with more female CEOs may yield different results. Hypothesis 2 was not supported, as CEO age had no significant impact on either return on assets (coefficient = 0.3038, $p > 0.05$), or return on equity (coefficient = 0.2035, $p > 0.05$). Hypothesis 3 was not supported, as CEO educational level had no significant effect on either measure of profitability (coefficient = 0.0966, $p > 0.05$, for return on assets; coefficient = 0.0821, $p > 0.05$, for return on equity). Hypothesis 4 was supported as CEO family membership had a significant positive reaction with return on assets (coefficient = 0.2352, $p < 0.001$) and a significant positive reaction with return on equity (coefficient = 0.2841, $p < 0.001$). Hypothesis 5 was supported, as CEO ownership of the family business had a significant positive reaction with return on assets (coefficient = 0.1034, $p < 0.001$) and a significant positive reaction with return on equity (coefficient = 0.1483, $p < 0.001$). Hypothesis 6 was supported contrary to the hypothesized direction, as CEO duality had a significant negative reaction with return on assets (coefficient = $-0.2091$, $p < 0.001$) and a significant negative reaction with return on equity (coefficient = $-0.2411$, $p < 0.001$). Hypothesis 7 was not supported as the number of CEO directorships had no effect on either measure of profitability (coefficient = 0.1393, $p > 0.05$ for return on assets; coefficient = 0.3284, $p > 0.05$, for return on equity). Hypothesis 8 was supported as CEO tenure had a significant positive reaction with return on assets (coefficient = 0.2008, $p < 0.001$) and a significant positive reaction with return on equity (coefficient = 0.2503, $p < 0.001$).

**Table 3.** Correlation matrix of dependent variables, independent variables, and control variables.

| Symbol | ROA | ROE | GEN | AGE | EDU | FAM | OWN | DUA | DIR | TEN | FAGE | FSIZE |
|---|---|---|---|---|---|---|---|---|---|---|---|---|
| ROA | 1 | | | | | | | | | | | |
| ROE | 0.7932 | 1 | | | | | | | | | | |
| GEN | 0.0731 | 0.0841 | 1 | | | | | | | | | |
| AGE | 0.1645 | 0.1849 | 0.0028 | 1 | | | | | | | | |
| EDU | 0.2019 | 0.2471 | 0.0311 | 0.0498 | 1 | | | | | | | |
| FAM | 0.1110 * | 0.0948 * | 0.0484 | 0.0595 | 0.0034 | 1 | | | | | | |
| OWN | 0.2018 * | 0.3191 * | 0.0131 | 0.0857 | 0.0492 | 0.0052 | 1 | | | | | |
| DUA | 0.2916 | 0.2947 | 0.0228 | 0.0020 | 0.0038 | 0.0022 | 0.0047 | 1 | | | | |
| DIR | 0.0448 | 0.0383 | 0.0411 | 0.0596 | 0.0104 | 0.0048 | 0.0484 | 0.0303 | 1 | | | |
| TEN | 0.0317 * | 0.0417 * | 0.0055 | 0.3052 ** | 0.0048 | 0.0551 | 0.0946 | 0.0455 | 0.0485 | 1 | | |
| FAGE | 0.1526 | 0.2049 | 0.0032 | 0.0394 | 0.0028 | 0.0494 | 0.0847 | 0.0232 | 0.0229 | 0.4058 | 1 | |
| FSIZE | 0.0193 | 0.0817 | 0.0485 | 0.0491 | 0.0585 | 0.0041 | 0.0037 | 0.0492 | 0.0558 | 0.0471 | 0.0145 | 1 |

\* $p < 0.05$, \*\* $p < 0.01$.

**Table 4.** Regression of return on assets (ROA) and return on equity (ROE) on CEO characteristics. Results of regressions of CEO personal characteristics and CEO corporate governance-related characteristics on firm performance.

| | ROA | | | | ROE | | | |
|---|---|---|---|---|---|---|---|---|
| | Coefficient | Std. Error | t | Significance | Coefficient | Std. Error | T | Significance |
| (Constant) | 1.7948 *** | 0.3084 | 5.8197 | 0.0002 | 1.6891 | 0.2262 | 7.4673 | 0.0000 |
| Gender | 0.1034 | 0.0854 | 1.2108 | 0.2538 | 0.1504 | 0.0994 | 1.5131 | 0.1612 |
| Age | 0.3038 | 0.2111 | 1.4391 | 0.1807 | 0.2035 | 0.1429 | 1.4241 | 0.1849 |
| Education | 0.0966 | 0.0564 | 1.7128 | 0.1175 | 0.0821 | 0.0615 | 1.3350 | 0.2115 |
| Family Membership | 0.2352 *** | 0.0438 | 5.3699 | 0.0003 | 0.2841 *** | 0.0384 | 7.3984 | 0.0000 |
| Ownership | 0.1034 *** | 0.0251 | 4.1195 | 0.0021 | 0.1483 *** | 0.0162 | 9.1543 | 0.0000 |
| Tenure | 0.2008 *** | 0.0371 | 5.4124 | 0.0003 | 0.2503 *** | 0.0215 | 11.6419 | 0.0000 |
| Directorships | 0.1393 | 0.0764 | 1.8233 | 0.0982 | 0.3284 | 0.1852 | 1.7732 | 0.1066 |
| CEO Duality | −0.2091 *** | 0.0229 | −9.1310 | 0.0000 | −0.2411 *** | 0.0133 | −18.1278 | 0.0000 |
| Firm Age | 0.1184 | 0.0948 | 1.2489 | 0.2401 | 0.1561 | 0.0847 | 1.8430 | 0.0951 |
| Firm Size | 0.1618 | 0.0791 | 2.0455 | 0.0680 | 0.2027 | 0.0972 | 2.0854 | 0.0636 |
| R square | 0.5083 | | | | 0.4892 | | | |
| Adjusted R square | 0.5019 | | | | 0.4277 | | | |

\*\*\* $p < 0.001$.

As a robustness check, both regressions specified in Equations (1) and (2) were subjected to GMM estimation and Two-Stage Least Squares, as shown in Table 4. All results from the OLS regression were supported, suggesting that problems of endogeneity of independent variables, the impact of unobserved variables, and the lack of quality of the findings did not exist.

The full sample was split into sub-samples of French firms and all other firms, as French firms constitute 77% of the total sample and may bias the full sample results. As shown in Tables 5 and 6, biases may only exist for CEO ownership and CEO tenure, as other countries did not exhibit significant effects on profitability of these two predictors. A supplementary analysis was carried out with Firm Year and Industry as control variables. Results remained unchanged, as shown in Table 7.

**Table 5.** Robustness tests of CEO characteristics on firm performance using Generalized Method of Moments and Two-Stage Least Squares GMM Model and Two-Stage Least Squares Model.

| | GMM Model | | 2SLS Regression | |
|---|---|---|---|---|
| | ROA | ROE | ROA | ROE |
| Gender | 0.1047 [ns] | 0.1504 [ns] | 0.1033 [ns] | 0.1456 [ns] |
| Age | 0.3033 [ns] | 0.2014 [ns] | 0.3057 [ns] | 0.2102 [ns] |
| Education | 0.0961 [ns] | 0.0844 [ns] | 0.0951 [ns] | 0.0783 [ns] |

**Table 5.** *Cont.*

|  | GMM Model | | 2SLS Regression | |
| --- | --- | --- | --- | --- |
|  | ROA | ROE | ROA | ROE |
| Family Member Ship | 0.2344 *** | 0.2841 *** | 0.2321 *** | 0.2756 *** |
| Ownership | 0.1052 ** | 0.1482 ** | 0.1052 ** | 0.1375 *** |
| Tenure | 0.2022 *** | 0.2503 *** | 0.2051 *** | 0.2427 *** |
| Director Ships | 0.1393 ns | 0.3211 ns | 0.1346 ns | 0.3049 ns |
| Duality | −0.2031 *** | −0.2411 *** | −0.2033 *** | −0.2392 *** |
| Firm Age | 0.1133 ns | 0.1544 ns | 0.1152 ns | 0.1601 ns |
| Firm Size | 0.1613 ns | 0.2034 ns | 0.1655 ns | 0.2144 ns |
| Wald Test | 14.5223 | 15.2231 | | |
| Hansen Test | 0.6242 | 0.6193 | | |
| $R^2$ | | | 0.5127 | 0.5531 |
| Adjusted $R^2$ | | | 0.4574 | 0.5091 |
| N | 137 | 137 | 137 | 137 |

** $p < 0.01$, *** $p < 0.001$, ns = not significant.

**Table 6.** Regression model of sub-samples regressions of CEO personal characteristics and CEO corporate governance mechanisms on firm performance for sub-samples of Euronext firms.

|  | France | | Other Countries | |
| --- | --- | --- | --- | --- |
|  | ROA | ROE | ROA | ROE |
| Gender | 0.1192 ns | 0.1612 ns | 0.0909 ns | 0.1205 ns |
| Age | 0.3392 ns | 0.2014 ns | 0.2837 ns | 0.2007 ns |
| Education | 0.1038 ns | 0.0844 ns | 0.0847 ns | 0.0641 ns |
| Family member Ship | 0.2411 *** | 0.2841 *** | 0.1984 *** | 0.2202 *** |
| Ownership | 0.1242 ** | 0.1482 ** | 0.0966 | 0.1375 |
| Tenure | 0.2494 *** | 0.2503 *** | 0.1891 | 0.2012 |
| Director Ships | 0.1491 ns | 0.3211 ns | 0.1131 ns | 0.3049 ns |
| Duality | −0.2241 *** | −0.2411 *** | −0.1895 * | −0.2114 * |
| Firm Age | 0.1109 ns | 0.1544 ns | 0.1002 ns | 0.1482 ns |
| Firm Size | 0.14833 ns | 0.2034 ns | 0.1348 ns | 0.2015 ns |
| $R^2$ | 0.6047 | 0.5283 | 0.6183 | 0.6513 |
| Adjusted $R^2$ | 0.5123 | 0.4742 | 0.5221 | 0.5627 |

* $p < 0.05$, * * $p < 0.01$, *** $p < 0.001$, ns = not significant.

**Table 7.** Regression of return on assets (ROA), and return on equity (ROE), on CEO characteristics including industry and year as control variables. Results of regressions of CEO personal characteristics and CEO corporate governance-related characteristics on firm performance.

|  | ROA | | | | ROE | | | |
| --- | --- | --- | --- | --- | --- | --- | --- | --- |
|  | Coefficient | Std. Error | t | Significance | Coefficient | Std. Error | T | Significance |
| (Constant) | 1.7423 *** | 0.3073 | 5.8122 | 0.0000 | 1.7101 *** | 0.2411 | 6.7011 | 0.0000 |
| Gender | 0.1041 | 0.0834 | 1.2152 | 0.3212 | 0.1201 | 0.0975 | 1.3411 | 0.1311 |
| Age | 0.3042 | 0.2155 | 1.4373 | 0.2521 | 0.3135 | 0.1311 | 1.5341 | 0.1671 |
| Education | 0.1055 | 0.0532 | 1.7163 | 0.1347 | 0.0762 | 0.0742 | 1.2333 | 0.2331 |
| Family Membership | 0.1944 *** | 0.0433 | 5.3622 | 0.0000 | 0.2611 *** | 0.0333 | 7.2144 | 0.0000 |
| Ownership | 0.1213 *** | 0.0256 | 4.1321 | 0.0032 | 0.1317 *** | 0.0156 | 8.1543 | 0.0000 |
| Tenure | 0.2022 *** | 0.0372 | 5.6221 | 0.0000 | 0.2411 *** | 0.0242 | 12.644 | 0.0000 |
| Directorships | 0.1352 | 0.07622 | 1.3351 | 0.0651 | 0.2141 | 0.1938 | 1.6332 | 0.1103 |
| CEO Duality | −0.1844 *** | 0.0266 | −8.6262 | 0.0001 | −0.2333 *** | 0.0121 | −17.1422 | 0.0000 |
| Firm Age | 0.1152 | 0.0963 | 1.5225 | 0.3310 | 0.1151 | 0.0784 | 1.8342 | 0.0622 |
| Firm Size | 0.1555 | 0.0733 | 1.9545 | 0.0731 | 0.2931 | 0.0525 | 2.0552 | 0.0791 |
| Year | | | | | | | | |
| Industy | | | | | | | | |

**Table 7.** *Cont.*

| | ROA | | | | ROE | | | |
|---|---|---|---|---|---|---|---|---|
| | Coefficient | Std. Error | t | Significance | Coefficient | Std. Error | T | Significance |
| R square | | 0.5083 | | | | 0.4892 | | |
| Adjusted R square | | 0.5019 | | | | 0.4277 | | |
| N | | 137 | | | | 137 | | |

*** $p < 0.001$.

## 5. Discussion

### 5.1. Discussion of Results

#### 5.1.1. Results in the Context of the Theoretical Framework

Upper Echelon Theory maintains that personal characteristics such as CEO age, gender, education, and nationality create a knowledge base of experiences and values that influence their perception of corporate situations. Family membership was the only personal characteristic that increased return on assets and return on equity, suggesting that CEOs who are family members have privileged knowledge of family likes and dislikes that enables them to win support from family members for the strategies and policies adopted by the family business. Winning such support is crucial as family members with large equity stakes in the business have the power to prevent the launching of new products, expansion into new markets, forming productive business partnerships, and other growth-enhancing strategies.

Agency theory maintains that CEOs may employ strategies that derive personal benefit to the detriment of firm profitability. This study found that CEO ownership reduces agency conflicts by increasing return on assets and return on equity. CEOs who own large amounts of equity in Euronext family businesses may demand accountability from management in order to safeguard the value of their own investment. They will require managers to adopt growth-enhancing strategies that increase return on assets and return on equity. An opposing effect on firm profitability is realized by CEO duality. CEOs who are Board Chairs may prevent the implementation of profit-oriented measures if they use idle cash to pay for strategies that give themselves visibility and praise even though such strategies may reduce firm profitability. An example would be international expansion that fails to yield increases in market share, but makes the CEO appear forward-looking.

Resource Dependency Theory maintains that CEO skills and experiences be matched to the resources and needs of the organization. CEO tenure was found to increase return on assets and increase return on equity. In family businesses, long-tenured CEOs become proficient in strengthening relationships between vendors, industry groups, CEOs of other firms, and family members. Vendors provide supplies, industry groups assist with regulation, and other CEOs may become future business partners. The matching of interpersonal skills of CEOs to these family business needs is an application of Resource Dependency Theory to family businesses.

#### 5.1.2. CEO Personal Characteristics and Firm Performance

Family Membership: Family membership was the CEO characteristic that improved financial performance in all samples, i.e., the full sample, and both sub samples. In contrast, CEO duality decreased return on assets and decreased return on equity in all samples. These findings suggest that family CEOs develop longstanding relationships with family members, clients, employees, and other stakeholders, which results in the smoothing of obstacles to implementation of strategies, leading to increased profitability. Resistance to implementation of strategies disappears as other stakeholders unite with the CEO in striving for superior financial performance. As the CEO is a family member, he or she has a personal relationship with the rest of the family. This is particularly useful, as the support of influential family members is essential to implement a strategic direction for the firm, as they can win the cooperation of other family members with voting privileges.

Stewardship theory underlies the connection between the CEO and other stakeholders. As a steward of the firm, the CEO displays a strong commitment to maintaining the firm's

reputation. In support of Adams et al.'s (2009) thesis, CEOs who are family members display an emotional connection to the business. The employee engagement literature suggests that employees (in this case, CEOs), manifest engagement with their jobs by displaying emotional attachment to their work that contributes to task performance (proficiency in task completion) and organizational citizenship (volunteering to assist peers) (Stewart and Brown 2011). The striving for an emotionally connected CEO may lead to improved financial performance.

CEOs are concerned about their status within the family, as poor financial performance downgrades the CEOs position in the family (Adams et al. 2009). The CEO may have high inherited status, as the offspring of the founder. Alternatively, the CEO may be the offspring of a low-status person, who has achieved high status by creating wealth for the family through creative entrepreneurship. In either case, the CEO wishes to maintain a high status within the family. As an example, in a manufacturer of rubber products, the family terminated the employment of a family CEO selected by the founder, who failed to achieve profit targets, replacing her with another family member with clearly developed financial goals. Therefore, family member CEOs are conscious of their need to support the livelihood of the family if they wish to maintain the high status that being a CEO confers upon them.

### 5.1.3. Corporate Governance Mechanisms Which Influence CEO Performance

CEO Duality: CEO duality significantly decreases profitability, suggesting that although stakeholders appreciate loyalty and family relationships with a CEO who is a member of the family, they do not wish to have a CEO who serves in the dual capacity of Board Chair. The underlying causes may be two-fold. Stakeholders may feel comfortable with a family CEO with whom they have a strong relationship; however, they may be wary of the concentration of power that a CEO who is Board Chair will possess. As Board Chair, the CEO is subject to the agency conflict of having weakened board control of management (Ujunwa 2015). In other words, CEOs of family businesses listed in the Euronext increase profitability by virtue of increased cooperation from key stakeholders, yet those same stakeholders wish to place limits on CEO power by preventing the CEO from being partial to other members of the top management team.

CEO Ownership Concentration, and CEO Tenure: We find that CEO ownership of stock and CEO tenure increased profitability. This finding supports stakeholder theory, as deeper CEO involvement with the firm through stock ownership and longevity results in stronger feelings of stewardship, and in turn, striving to achieve higher profits. CEOs who are major shareholders care about firm performance, as it impacts the value of their stock. The longer they are employed, the better they will be able to monitor the firm's performance, deploying measures to overcome underperformance due to failed policies, or macroeconomic conditions, such as inflation and stagnation.

### 5.2. Hypotheses That Were Not Supported and Control Variables

Demographic characteristics such as age, gender, and education together with the corporate governance mechanism of number of directorships were not significant predictors of return on assets and return on equity. We may consider conditions under which these hypotheses may be supported. This information may be used to conduct future studies. For gender, a larger sample with similar numbers of male and female CEOs would provide observations that measure gender effects more accurately, as there are more women than men in our sample who could have an effect on firm performance. Likewise, for age, the inclusion of younger CEOs may provide the breadth of sample respondents to truly assess the impact of age, as the mean age of the CEOs in this study is 55 years. For education, the mean < 1 may indicate low levels of college education of a bachelor's degree or less. Perhaps, three levels of education with two levels of graduate degrees may yield different results. Directorships seem to be adequate with 2.5 as the mean. Maybe family businesses in the Euronext region do not need the diversity of experiences that exposure to multiple

firms through directorships confer upon the CEO, due to the high similarity of experiences for board members across firms.

Firm Age showed a difference in significance between ROA as the criterion and ROE. In the regressions, ROE showed higher significance than ROA. Older firms may have had higher return on equity as they are manufacturing firms with smaller equity bases and higher cash flows. Conversely, younger firms may be service firms with larger equity bases.

## 6. Conclusions

### 6.1. Theoretical Contributions

This study adds to the existing body of knowledge on the impact of CEO characteristics on the financial performance of family businesses. This paper makes four specific contributions to the knowledge. The results for family businesses in the Euronext region closely match the results obtained for family businesses and nonfamily businesses in other regions. First, family membership has been found to increase return on assets, return on equity, and Tobin's q in prior studies (Adams et al. 2009). In this prior study, founder CEOs were found to have a large positive influence on firm performance. As a member of the family that started the business, the individual who was the principal founder assumes personal responsibility for the success of the business. We consider such feelings of personal responsibility and stewardship to drive the positive effects on profitability of CEOs in our sample. Second, ownership concentration was found to reduce agency conflicts in the prior studies of El-Chaarani et al.'s (2022) examination of bank boards in the MENA region and El-Chaarani et al.'s (2023) study of bank boards in Gulf Cooperation Council countries. These institutions were non-family businesses so that this study's similar finding of positive effects extends their finding to family businesses. Further, these studies did not specifically isolate the effects of CEO ownership, as they included board members. This study extends the finding that ownership concentration among board members results in higher return on assets and return on equity to CEOs of family businesses. Third, CEO duality with adverse effects on financial performance in both the Palaniappan (2017) and Pandey et al. (2023) samples using Indian data can be extended to family businesses in the Euronext region. Finally, in both the Francis et al. (2008) study and Ali and Zhang's (2015) examination of non-family businesses, CEO tenure increased firm profitability, as it did in this study of family businesses. Once again, a result for non-family businesses may be extended to family businesses.

### 6.2. Contributions to the Existing Literature

This study supports the existing findings of family membership improving financial performance, and CEO duality adversely affecting financial performance (see Cucculelli et al.'s (2019) examination of seven European family firms and Garcia-Ramos and Garcia-Olalla's (2014) finding of the negative effects of CEO duality in a predominantly non-Euronext sample). Our results suggest that there is some similarity in the effects of CEO characteristics on firm performance between family firms listed in the Euronext and those in the rest of Europe. We add to this nascent literature on European family businesses, with the findings of CEO ownership and CEO tenure improving financial performance in the French sample. Future research could be qualitative in nature, with interviews with French CEOs to determine the aspects of CEO ownership and aspects of CEO tenure that positively influence financial performance. Certain research questions could be addressed through interviews. Does ownership have cultural roots that increase profitability? Does tenure provide the job security to stimulate the adoption of creative solutions to problems?

There is lack of support for key demographic variables, such as age, gender, and education, in contrast with the strong support for corporate governance mechanisms, including CEO ownership, CEO duality, and CEO tenure, influencing financial performance. Perhaps, the creation of an environment of transparency, and accountability, is more important in family firms than any personal attributes of the CEO.

### 6.3. Practical Implications

The principal value of this study for practice lies in its addition to the criteria for hiring CEOs in family businesses. Family membership and separation of the CEO position from that of the Board Chair is advised for family firms listed in the Euronext, as such firms are likely to earn higher return on assets and return on equity. In French firms, financial performance will be enhanced further if the CEO owns stock in the firm and is long-tenured.

Family firms frequently seek additional sources of capital. Firms with family members in the position of CEO and those with separation of the CEO from the Board Chair may be in a position to raise capital from new sources, as lenders will be impressed by their increased profitability. Conversely, firms with CEO duality may convey a message of managerial entrenchment. Managerial entrenchment suggests weakened Board control of management or a lack of accountability for financial results by management, as they are being evaluated by a Board Chair who is a member of their own team. As such duality is shown in this study to diminish profitability, investors will be reluctant to invest in such a firm, inhibiting the firm's ability to raise capital.

### 6.4. Limitations of the Study

This study was performed over a short two-year period. Future research should be conducted over a longer period to capture any predictors that have emerged at other points in time. Return on assets and return on equity could be supplemented by Tobin's Q and a market to book ratio to provide a richer measure of financial performance. Bias toward the French sample may be eliminated by increasing the sample size with non-French firms. Quantitative results may be enriched with CEO interviews and CEO surveys, or qualitative measures of CEO characteristics. This study may be repeated with larger numbers of professional managers as CEOs who are non-family members to determine the influence of such individuals on firm performance. Comparisons may be performed between family member CEOs and non-family member CEOs who are professional managers.

**Author Contributions:** Conceptualization, Z.E.A.; methodology, H.E.-C.; formal analysis, Z.E.A.; investigation, R.O.B.; resources, R.A.; data curation, S.H.J.; writing original draft, H.E.-C. and R.A.; writing review and editing, R.A.; visualization, R.A. and R.O.B.; supervision, R.O.B.; project administration, Y.S. and S.H.J. All authors have read and agreed to the published version of the manuscript.

**Funding:** This research received no external funding.

**Data Availability Statement:** Data may be obtained from the third author upon request.

**Conflicts of Interest:** The authors declare no conflicts of interest.

### Appendix A

**Table A1.** List of firms employed in this sample.

| Firm Name | Company Code | Country of Location |
| --- | --- | --- |
| Ab inbev | BE0974293251 | Leuven, Belgium |
| Cie bois sauvage | BE0003592038 | Bruxelles, Belgium |
| Miko | BE0003731453 | Turnhout, Belgium |
| Nextensa | BE0003770840 | Brussels, Belgium |
| Recticel | BE0003656676 | Brussels, Belgium |
| SIPEF | BE0003898187 | Scholen, Belgium |
| Sofina | BE0003717312 | Etterbeek, Belgium |
| Solvay | BE0003470755 | Brussels, Belgium |
| Texaf | BE0974263924 | Brussels, Belgium |
| UCB | BE0003739530 | Brussels, Belgium |
| Umicore | BE0974320526 | Brussels, Belgium |
| Advini | FR0000053043 | Saint-Felix-de-Lodz, France |
| Akwel | FR0000053027 | Champtromier, France |
| Axway | FR0011040500 | Puteaux, France |
| Bassac | FR0004023208 | Cognac, France |

**Table A1.** *Cont.*

| Firm Name | Company Code | Country of Location |
|---|---|---|
| Beneteau | FR0000035164 | Saint-Gilles-Croix-de-Vie, France |
| Biocorp | FR0012788065 | Issoire, France |
| Biomerieux | FR0013280286 | Saint-Vulbas, France |
| Boiron | FR0000061129 | Aubiere, France |
| Bollore | FR0000039299 | Puteaux, France |
| Bonduelle | FR0000063935 | Villeneuve d'Ascq, France |
| Bouygues | FR0000120503 | Paris, France |
| Catering intl sces | FR0000064446 | Marseille, France |
| Cegedim | FR0000053506 | Amilly, France |
| Christian dior | FR0000130403 | Paris, France |
| Clasquin | FR0004152882 | Lyon, France |
| Damartex | FR0000185423 | Roubaix, France |
| Dassault systemes | FR0014003TT8 | Paris, France |
| Delta plus group | FR0013283108 | Apt, France |
| Derichebourg | FR0000053381 | Paris, France |
| Exacompta clairef. | FR0000064164 | Etival Clairefontaine, France |
| Fleury michon | FR0000074759 | Plan-les-ouates, France |
| Fountaine pajot | FR0010485268 | Aigreteuille d'Aunis, France |
| Gpe group pizzorno | FR0010214064 | Draguignan, France |
| Groupe crit | FR0000036675 | Paris, France |
| Groupe guillin | FR0012819381 | Beausemblant, France |
| Guerbet | FR0000032526 | Villepinte, France |
| Herige | FR0000066540 | L'Herbergement, France |
| Hermes intl | FR0000052292 | Paris, France |
| Immob.dassault | FR0000033243 | Paris, France |
| Installux | FR0000060451 | Saint Bonnet de Mure, France |
| Ipsen | FR0010259150 | Boulon, Billiancourt, France |
| Jacques bogart | FR0012872141 | Paris, France |
| Kering | FR0000121485 | Paris, France |
| L''oreal | FR0000120321 | Cruzier-le-Vieux, France |
| Lacroix group | FR0000066607 | Saint Herblain, France |
| Lanson-bcc | FR0004027068 | Reims, France |
| Laurent-perrier | FR0006864484 | Tours-sur-Marne, France |
| Les hotels baverez | FR0007080254 | Paris, France |
| Lna sante | FR0004170017 | Vertou, France |
| Lvmh | FR0000121014 | Paris, France |
| Manitou bf | FR0000038606 | Ancenis, France |
| Michelin | FR001400AJ45 | Saint Doulchard, France |
| Oeneo | FR0000052680 | Paris, France |
| Pernod ricard | FR0000120693 | Bordeaux, France |
| Piscines desjoyaux | FR0000061608 | Saint Maur, France |
| Precia | FR0014004EC4 | Occitania, France |
| Prodways | FR0012613610 | Mureaux, France |
| Saint jean groupe | FR0000060121 | Belleville en Beaujolais, France |
| Savencia | FR0000120107 | Viroflay, France |
| Sodexo | FR0000121220 | Courbevoie, France |
| Synergie | FR0000032658 | Paris, France |
| Tff group | FR0013295789 | Saint-Herblain, France |
| Valneva SE | FR0004056851 | Saint-Herblain, France |
| Vetoquinol SA | FR0004186856 | Lure cedex. France |
| Vivendi | FR0000127771 | Paris, France |
| Kingspan Group | IE0004927939 | Kingscourt, Ireland |
| ArcelorMittal | LU1598757687 | Luxembourg, Luxembourg |
| Heineken | NL0000009165 | Amsterdam, The Netherlands |
| Vastned | NL0000288918 | Amsterdam, The Netherlands |
| Jerónimo Martins SGPS, SA | PTJMT0AE0001 | Lisbon, Portugal |

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
