# Peer review of "The Impact of CEO Characteristics on the Financial Performance of Family Businesses Listed in the Euronext Exchange"

_jrfm, doi:10.3390/jrfm17030129_

Round 1

Reviewer 1 Report

Comments and Suggestions for Authors

See comments in the attached document. 

Comments on the Quality of English Language

Author Response

See items in yellow in the attached file.

Reviewer 2 Report

Comments and Suggestions for Authors

Title: Ok. does corresponds with the journal scope and paper content

Abstract: It does explain what is going on in the paper. However, the wording of the results presentation is not correct -> Even with all tests and robust results, being so sure that these characteristics impact the business performance is too optimistic. I would rather suggest to write those results suggest what there is evidence (or that there is indication) that i.e. "CEO family membership has positive impact on return on assets"... and so on.

Introduction: It shows that the authors are aware of the Family business domain and the CEO traits that influence firms’ performance. The problem is well established with relevant citation. I miss a clear goal statement. What is the purpose of this paper (to close the research gap or investigate a phenomenon?).

Literature review: This part discuses three concepts that were used to develop the hypotheses. I suggest to review the introduction part and lit. review since it describes similar problems. I would expect the introduction shorter than lit. review. In the review there should be a clear goal why and what for are you performing it? It should help to support the significance of your paper and the phenomena you are researching.

Section 3: I like the review behind the phenomena you are focusing on. So then what is the purpose of the previous review (maybe this section can a subsection of lit review)

Section 4: Written in understandable and clear way. Just to be sure in the Table 1 the ROA is a decimal number and ROE as well or its in percentage? (10.7 would seem like 1070% ROE). Also, I would suggest to use ROE only since it is better for inter industry comparisons. Beware that ROE is dependent on ROA? have you checked autocorrelation? Results of your study for both are quite similar.... ROA might be highly industry dependent. However, the control variable might negate this effect.

Few more suggestions: The subsection should not finish with a Table rather with text. Section 4.2 is not a proper text.. I would suggest to rewrite it into proper sentences or make a bullet point paragraph... Ownership-> means number of shares in what metric (number, or millions, thousands), firm size in thousands of EUR or mil EUR?

Small typo in the second formula.

Section 5: I am a bit sceptical on the results regarding gender, since your sample is small and does have a lot of male CEOs... there is potential for error.

Also I am not convinced by the wording you are using, since claims such as "CEO duality significantly decreasing return on assets".. similarly, as in the abstract, there is so many different and more influential characteristics that influence the ROE or ROA, that such claims sound far too unrealistic. Just stick to what you have found: the CEO duality may have a significant negative relation with return on assets, thus the duality may be one of the factors that negatively influence th firms performance... In this way your claim is less certain but closer to the truth.

So even good statistical results are not enough (also given the sample) to make strong claims (or maybe juts needed to be better written and explained)

Section 6 (Conclusion should be number 6??) I would put discussion and conclusion separately. The discussion subsection only comments on hypotheses that were supported. Ok but i would welcome some discussion about the results that have not supported the hypotheses. Also, you the same stratification of hypotheses from section 3 so that the reader is not confused.

The citations are in line of what you write, so that is correct.

Subsection 7.3 (6.3) Hmm, “Family membership and separation of the CEO from that of Board Chair is advised for family firms listed on the Euronext, as such firms are likely to earn higher return on assets, and return on equity”. (line 537-539) Well, in my opinion the relation can be opposite, since companies that are successful and growing mostly need more professional managers, so they are more attractive as hires... But definitely, if a family company is growing and has high profitability, to sustain, it needs separation of management and ownership. It is also quite attractive to owners to split with the company day to day business at that stage.

I agree that family owned firms tend to be more risk averse -> SO WHY did you only focus on ROE and ROA. Why not debt ratio or financial leverage, or liquidity???? Focusing only on two and possibly interrelated ratios is waste of your time in my opinion. Try to run it for different and less dependent financial performance measures...

I agree with limitations...Sure what you suggest should be performed to have more insight.

Author Response

See items in green for Reviewer 2 in the attached file.

Reviewer 3 Report

Comments and Suggestions for Authors

Review of “The Impact of CEO Characteristics on the Financial Performance of Family Businesses Listed on the Euronext Exchange

Manuscript ID: jrfm-2903642

Overall comments

I found the paper well written and the idea interesting.  The literature review is well done and extensive.  The discussion of results is good.

Issues:

1.         In the section 4.2 Definition of Variables the authors should include Return on Assets and Return on Equity and a discussion of how calculated.  In addition, I think that there is a mistake with the Gender variable. I think it should be 1 for men and 0 for women (not 2) as this is what the table 1 suggests.

2.         Section 4.3 there is a small typo on the second equation – it should be equity not "equit".

3.         I have two questions concerning the Firm Age variable.  First, the average does not seem reasonable if it is the log of number of years, 20.0912, I think you meant 2.00912. This should be checked.

Second, I wish there would have been additional discussion of the Firm Age finding that showed a difference when examining ROA and ROE.  This seems like an interesting finding as industry is not examined.  I think looking into this finding may prove interesting.  For example are older firms more likely to engage in manufacturing while younger are more service oriented?

Author Response

See items in blue in the attached file.

Reviewer 4 Report

Comments and Suggestions for Authors

The topic of the paper is very interesting and actual. The authors investigate if CEO characteristics, i.e. personal characteristics, and corporate governance mechanisms have an impact on the performance of family companies listed on the Euronext. However, the paper needs modifications:

1.       Line 16: Please replace “137 family firms” with “137 firm-year observations”

2.       I observed that the Authors did not include a correlation matrix table. I recommend presenting results on variable correlations. This information is crucial for a comprehensive understanding of the relationships within the study.

3.      The literature on family firms (see Sharma, P., Chrisman, J. J., & Chua, J. H. (1996). A review and annotated bibliography of family business studies. Norwell, MA: Kluwer Academic Publishers, p. 4-7; Klein S. (2000). Family Businesses in Germany: Significance and Structure. Family Business Review, 13(3), pp. 157-181, DOI: 10.1111/j.1741-6248.2000.00157.x) provides many definitions of family firms. Please explain how the family firms included in the research sample were defined.

4.      The study uses CEO duality as independent variable. This variable applies to the one-tier governance model. However, in some countries (e.g. France – see OECD Corporate Governance Factbook 2023) the company chooses between a one-tier or a two-tier board structure. Please explain if the research sample includes only companies with a one-tier board structure. 

5.       Table 1 shows two descriptive statistics: average and standard deviation. I recommend considering the inclusion of more detailed information on descriptive statistics (minimum, maximum, median).

6.       The research models seem incorrect. They include eight independent variables referring to CEO characteristics, and two control variables (age and size). The financial performance of companies is affected by many factors, thus other control variables might be included in regression models (i.e. growth, leverage, etc.). Please consider estimating more regression models but with a smaller number of independent variables referring to CEO characteristics and a greater number of control variables.

7.       Tables 2, 3 and 4 should include detailed information on the number of observations. Table 4 presents the results of regression analysis, but does the research sample is large enough?

8.       The discussion section might be improved.

9.       In my opinion, some parts of the paper (lines: 207-214; 242-249; 253-263; 312-332; 335-344; 351-370) need proper citations or some modification. The reader should know if the authors present their own opinions, and conclusions, or if they refer to the works of other authors.

Author Response

See items in purple in the attached file. 

Round 2

Reviewer 2 Report

Comments and Suggestions for Authors

I agree with perofrmend changes. After proper formating and some english corrections, the paper is sound.

Reviewer 4 Report

Comments and Suggestions for Authors

I found that the authors have taken a lot of effort to work on the review comments. Good luck!